# Forecasting of phenotypic and genetic outcomes of experimental evolution in *Pseudomonas protegens*

**Jennifer T. Pentz**, **Peter A. Lind** *

Department of Molecular Biology, Umeå University, Umeå, Sweden

* peter.lind@umu.se

**Data Availability Statement:** All relevant data are within the manuscript and its Supporting Information files. All Illumina sequencing files are available from the NCBI database (accession

## Abstract

Experimental evolution with microbes is often highly repeatable under identical conditions, suggesting the possibility to predict short-term evolution. However, it is not clear to what degree evolutionary forecasts can be extended to related species in non-identical environments, which would allow testing of general predictive models and fundamental biological assumptions. To develop an extended model system for evolutionary forecasting, we used previous data and models of the genotype-to-phenotype map from the wrinkly spreader system in *Pseudomonas fluorescens* SBW25 to make predictions of evolutionary outcomes on different biological levels for *Pseudomonas protegens* Pf-5. In addition to sequence divergence (78% amino acid and 81% nucleotide identity) for the genes targeted by mutations, these species also differ in the inability of Pf-5 to make cellulose, which is the main structural basis for the adaptive phenotype in SBW25. The experimental conditions were changed compared to the SBW25 system to test if forecasts were extendable to a non-identical environment. Forty-three mutants with increased ability to colonize the air-liquid interface were isolated, and the majority had reduced motility and was partly dependent on the Pel exopolysaccharide as a structural component. Most (38/43) mutations are expected to disrupt negative regulation of the same three diguanylate cyclases as in SBW25, with a smaller number of mutations in promoter regions, including an uncharacterized polysaccharide synthase operon. A mathematical model developed for SBW25 predicted the order of the three main pathways and the genes targeted by mutations, but differences in fitness between mutants and mutational biases also appear to influence outcomes. Mutated regions in proteins could be predicted in most cases (16/22), but parallelism at the nucleotide level was low and mutational hot spot sites were not conserved. This study demonstrates the potential of short-term evolutionary forecasting in experimental populations and provides testable predictions for evolutionary outcomes in other *Pseudomonas* species.

## Author summary

Biological evolution is often repeatable in the short-term suggesting the possibility of forecasting and controlling evolutionary outcomes. In addition to its fundamental importance

number(s) PRJNA737653, https://www.ncbi.nlm.
nih.gov/sra/PRJNA737653).

**Funding:** This work was supported by the Kempe
foundations (http://www.kempe.com/)(SMK-
1858.1)[P.A.L], Carl Trygger's Foundation for
Scientific Research (https://carltryggersstiftelse.se/)
(CTS 16:275) [P.A.L] and Magnus Bergvall's
Foundation (http://www.magnbergvallsstiftelse.nu/)
(2016)[P.A.L]. The funders had no role in study
design, data collection and analysis, decision to
publish, or preparation of the manuscript.

**Competing interests:** The authors have declared
that no competing interests exist.

for biology, evolutionary processes are at the core of several major societal problems,
including infectious diseases, cancer and adaptation to climate change. Experimental evo-
lution allows study of evolutionary processes in real time and seems like an ideal way to
test the predictability of evolution and our ability to make forecasts. However, lack of
model systems where forecasts can be extended to other species evolving under different
conditions has prevented studies that first predict evolutionary outcomes followed by
direct testing. We showed that a well-characterized bacterial experimental evolution sys-
tem, based on biofilm formation by *Pseudomonas fluorescens* at the surface of static
growth tubes, can be extended to the related species *Pseudomonas protegens*. We tested
evolutionary forecasts experimentally and showed that mutations mainly appear in the
predicted genes resulting in similar phenotypes. We also identified factors that we cannot
yet predict, such as variation in mutation rates and differences in fitness. Finally, we made
forecasts for other *Pseudomonas* species to be tested in future experiments.

## Introduction

An increasing number of experimental evolution studies, primarily using microbes, have pro-
vided insights into many fundamental questions in evolutionary biology including the repeat-
ability of evolutionary processes [1–6]. Given the ability to control environmental conditions,
population size, as well as the use of a single asexual organism, such studies could provide an
opportunity to predict evolutionary outcomes in simplified model systems. High repeatability
on both phenotypic and genetic levels have been observed in a large number of experimental
evolution studies (reviewed in [5]), but it has become clear that high repeatability alone is not
sufficient for testing evolutionary predictability beyond the prediction that under identical
conditions the same evolutionary outcome is probable.

The difficulties of moving from repeatability to predictability are largely a result of the lack
of knowledge of the genotype-phenotype-fitness map, including how sensitive it is to changes
in environmental conditions and to what degree it is conserved between different strains and
species [7–9]. Several problems arise when searching for suitable model systems for testing
and improving predictive ability (reviewed in detail in [10]): I. Adaptive mutations are often
strain specific, so that adaptation of different strains of the same species to an environmental
challenge, such as high temperature, often result in mutations in different genes. For example,
after adaptation to high temperature in minimal media with glucose for the *E. coli* strains K-12
MG1655 [11] and REL1206 [6], a *E. coli* B derivative, none of the top ten most frequently
mutated genes were shared between strains. II. The range of probable adaptive phenotypes can
often not be defined beforehand due to more than one dominant selective pressures [6, 12,
13], which means that in many cases the phenotypes that solve the focal selective problem are
outcompeted by other phenotypes with increased fitness. III. Adaptation to a specific selective
pressure may be solely explained by the molecular phenotype of a single protein, resulting in a
relatively simple parameter and genotype space, as is sometimes observed for high-level antibi-
otic resistance [14–19]. Thus, predictions will be identical for all species and the model system
cannot provide a test of predictions from general principles. The most useful model systems
for testing and developing our ability to predict evolution, at least at this point, are likely to be
of intermediate complexity with several beforehand recognizable phenotypic and genetic solu-
tions to combine ample opportunities for failure with a decent chance of success.

The wrinkly spreader (WS) model in *P. fluorescens* SBW25 (hereafter SBW25) is one of the
most well characterized experimental evolution model systems and has several suitable

properties, in relation to the problems described above, that could make it possible to extend knowledge and principles from this species to related species [20–27]. When the wild type SBW25 is placed into a static growth tube the oxygen in the medium is rapidly consumed by growing bacteria. However, the oxygen level at the surface is high and mutants with increased ability to colonize the air-liquid interface have a major growth advantage and rapidly increase in frequency (Fig 1A). Several phenotypic solutions to air-liquid interface colonization, all involving increased cell-cell adhesion, have been described and are distinguishable by their divergent colony morphologies on agar plates [20, 22, 25]. The most fit of these is the Wrinkly Spreader (WS) [20, 22] that overproduces cellulose that is the main structural component of the mat at the air-liquid interface [24, 28]. The WS phenotype is caused by mutational activation of c-di-GMP production by a diguanylate cyclase (DGC) [27]. The second messenger c-di-GMP is a highly conserved signal for biofilm formation in bacteria that controls formation of a matrix composed of polysaccharides, DNA and proteins, and reduce motility [29, 30]. While many different DGCs can be activated to produce the WS phenotype, some are greatly overrepresented due to larger mutational target sizes leading to a hierarchy of genetic routes to WS (Fig 1C) [21, 23]. The genotype-to-phenotype map to WS has been characterized in detail [23, 26, 27] allowing the development of mathematical models of the three main pathways to WS (Wsp, Aws and Mws) and the prediction of evolutionary outcomes [26].

The WS model could be extendable to other *Pseudomonas* strains or species if it can be shown that mutations are not mainly strain-specific and the selective pressure in terms of colonization of the air-liquid interface is dominant and robust to minor changes in environmental conditions. To put it simply, this means that closely related strains should, to some degree, evolve similar adaptive phenotypes by similar mutations in similar environments. An extended WS model would be ideal for testing evolutionary predictability because it is easy to isolate many clonal mutants using an initial step of screening for differences in colony morphology, characteristic of many mutants with increased ability to colonize the air-liquid interface in SBW25 [20, 22, 25]. This allows detection of mutants at <1% frequency in a population and reduces DNA sequencing costs. However, it also means that other types of adaptive mutants, including those adapting to other selective pressures, without divergent colony morphology are not included even if they reach high frequencies in the population. Strong selection allows short-term experiments where only one or two mutations are typically responsible for the adaptive air-liquid interface colonization phenotype and very few secondary mutations are typically present, which makes it straightforward to connect genotypes to phenotypes and fitness. The short timescale used, typically tens of generations, will however limit insights on the predictability of longer evolutionary trajectories.

Despite the reductionistic appeal of the WS system, there is considerable complexity both in the range of different phenotypic solutions to colonization of the air-liquid interface and in genetic pathways to the high fitness WS phenotype [20–23, 26, 31]. Moreover, this complexity would be even greater if it can be extended to other species as both DGCs and biofilm structural components, e.g. polysaccharides, are frequently gained by horizontal gene transfer and lost by deletions leaving a mosaic pattern of presence and absence that is not strictly linked to phylogeny. In many cases these genes are not essential, meaning that they can be deleted to allow finding alternative adaptive pathways [21, 22]. Detailed genetic and phenotypic data on adaptive mutants from experimental evolution under static conditions is available almost exclusively from SBW25, which is currently a major limitation. If the WS system can be extended, it would be possible to iteratively improve models and update predictions based on new experimental data to be less dependent on the idiosyncrasies of an individual strain and environment combination.

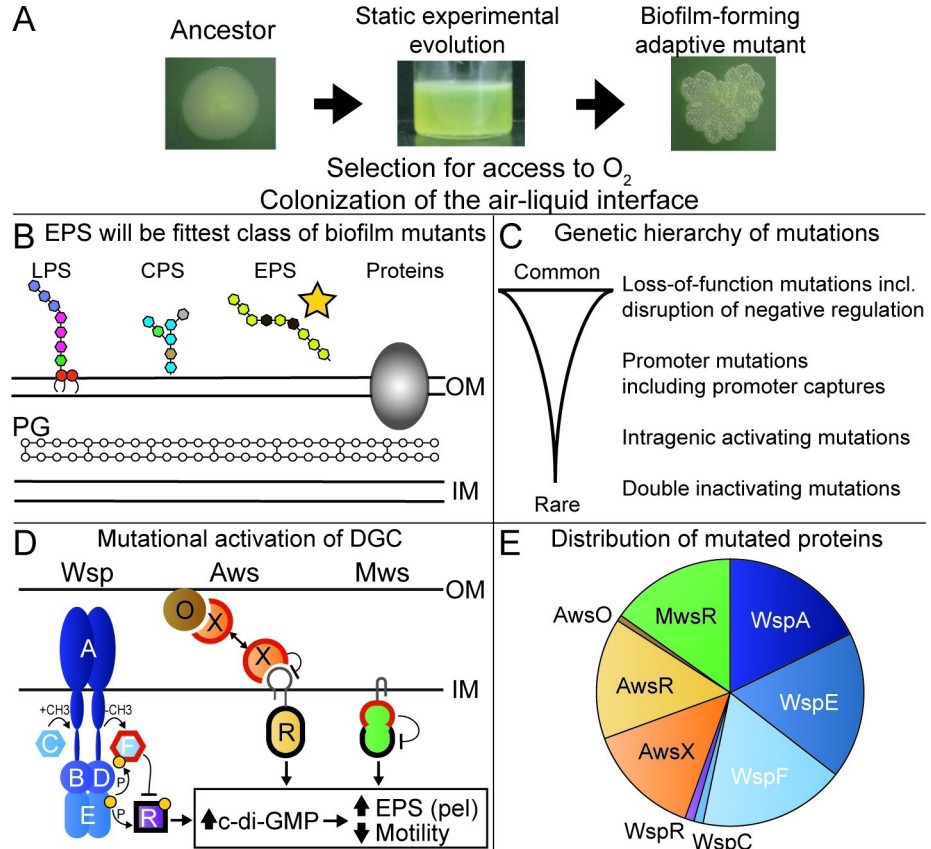

**Fig 1. Evolutionary predictions and outcomes of experimental evolution for *Pseudomonas protegens* Pf-5.** (*A*) Growth under static conditions when oxygen is limiting to growth is expected to result in selection for colonization of the air-liquid interface by increased cell-cell adhesion and surface attachment. Pictured is a wrinkly spreader mutant of *Pseudomonas fluorescens* SBW25. (*B*) The cell wall of Pf-5 has several components that could possibly be used to promote cell-cell adhesion and surface attachment, including lipopolysaccharides (LPS), capsular polysaccharides (CPSs), exopolysaccharides (EPSs), adhesive proteins, and incomplete cleavage of the peptidoglycan (PG) layer. (*C*) The types of adaptive mutations expected are, in decreasing frequency, loss-of-function mutations, promoter mutations, intragenic activating mutations, and double inactivating mutations [28]. (*D*) Mutational activation of di-guanylate cyclases (DGCs–outlined in black; WspR, AwsR, and MwsR) by loss-of-function mutations in negative regulators (outlined in red; WspF, AwsX, and MwsR) will lead to increased c-di-GMP production, resulting in overexpression of an EPS and reduced motility. In SBW25, these occur in three main pathways, Wsp, Aws, and Mws. (*E*) Predicted fraction of WS mutations for genes in the main three pathways. Knowledge of the molecular networks allowed for the formulation of a mathematical model that accurately predicted the relative rates of use of the common three pathways (Wsp, Aws, and Mws) [26] and rates for proteins in each pathway. The assumption that the molecular functions of these networks are conserved in Pf-5 allows prediction of the rates of pathways and proteins.

In this study, we use previous knowledge of the SBW25 WS system to make predictions of phenotypic and genetic evolutionary outcomes of experimental evolution for *Pseudomonas protegens* Pf-5 (hereafter Pf-5) that are then tested experimentally. Pf-5 has a highly conserved genetic repertoire of DGCs, including WspR, AwsR and MwsR that are most commonly activated by mutations in SBW25, but no experimental data on its response to experimental evolution under static conditions are available. A major difference between the species is that Pf-5 lacks genes for biosynthesis of the main structural component, cellulose, used for air-liquid interface colonization by WS types in SBW25. Results show that phenotypes, order of pathways used, and types of mutations can be predicted and that forecasts are robust to introduced changes in experimental conditions compared with the SBW25 system. Our results suggest

that there is potential for forecasting short-term evolutionary processes in an extended *Pseudomonas* wrinkly spreader system, and we conclude by making forecasts for five other species of *Pseudomonas* based mainly on their genome sequence and data from Pf-5 and SBW25 that will be tested in future studies.

## Results

### Predictions for experimental evolution of *Pseudomonas protegens* Pf-5

Our predictions are based on previous experimental data from SBW25 and *P. aeruginosa* and theoretical considerations that are briefly presented here together with supporting references and explained in detail in the S1 Text. We believe that forecasts of experimental evolution under novel conditions, such as for a new species and a new environment, need to start with identifying the dominant selective pressures that determines which phenotypes will have increased fitness before making genetic predictions. Each level will be dependent on correct higher-level predictions (except for prediction 4) meaning that, for example, if no mutants with increased ability to colonize the air-liquid interface evolve, we have no reason to expect that mutations would be found the predicted genes.

1. Mutants with increased ability to colonize the air-liquid interface by increased cell-cell or cell-wall adhesion will evolve and rise to high frequencies as these will be among the most fit classes of single-step adaptive mutants (Fig 1A) [20–22].

2. Mutants that use exopolysaccharides will have higher fitness than alternative phenotypic solutions with increased air-liquid interface colonization, such as lipopolysaccharides, capsular polysaccharides, cell-chaining or adhesive proteins (Fig 1B). As Pf-5 does not have genes required for biosynthesis of cellulose it is predicted to instead use the Pel exopolysaccharide [32].

3. The most fit class of mutants with increased ability to colonize the air-liquid interface will have reduced motility due to activation of DGCs resulting in increased c-di-GMP and exopolysaccharide production, and selection against the cost of motility in biofilms (Fig 1D) [21, 22, 30, 33–36].

4. Most mutations will cause loss of function, followed by promoter mutations and even less frequent activating mutations and double inactivating mutations (Fig 1C) [20–23, 37].

5. Mutations in the molecular networks of the negatively regulated DGCs WspR, AwsR and MwsR will be the most common route to WS due to a large mutational target size (Fig 1D) [21, 23, 26].

6. A previously developed mathematical model predicts that most mutations will be in WspA, WspE, WspF, AwsR, AwsX and MwsR [26] (Fig 1E). This prediction assumes that the functional interactions of the molecular networks are conserved between SBW25 and Pf-5 and that inactivating mutations are much more common than activating mutations. It does not assume that mutational hot spot sites are conserved between SBW25 and Pf-5.

7. Mutated regions for proteins in the Wsp, Aws and Mws pathways are predicted based on previous mutations in SBW25 [22, 23, 26, 27, 38] and *P. aeruginosa* [39–41] combined with analysis of homology models of protein structures [42] (Fig 2).

8. Mutations will not be evenly spread between predicted Wsp, Aws, Mws proteins with large mutational targets, but found mainly in those where mutations have the largest beneficial fitness effects [21, 23, 26]. This is because mutations with similar functional effects on a

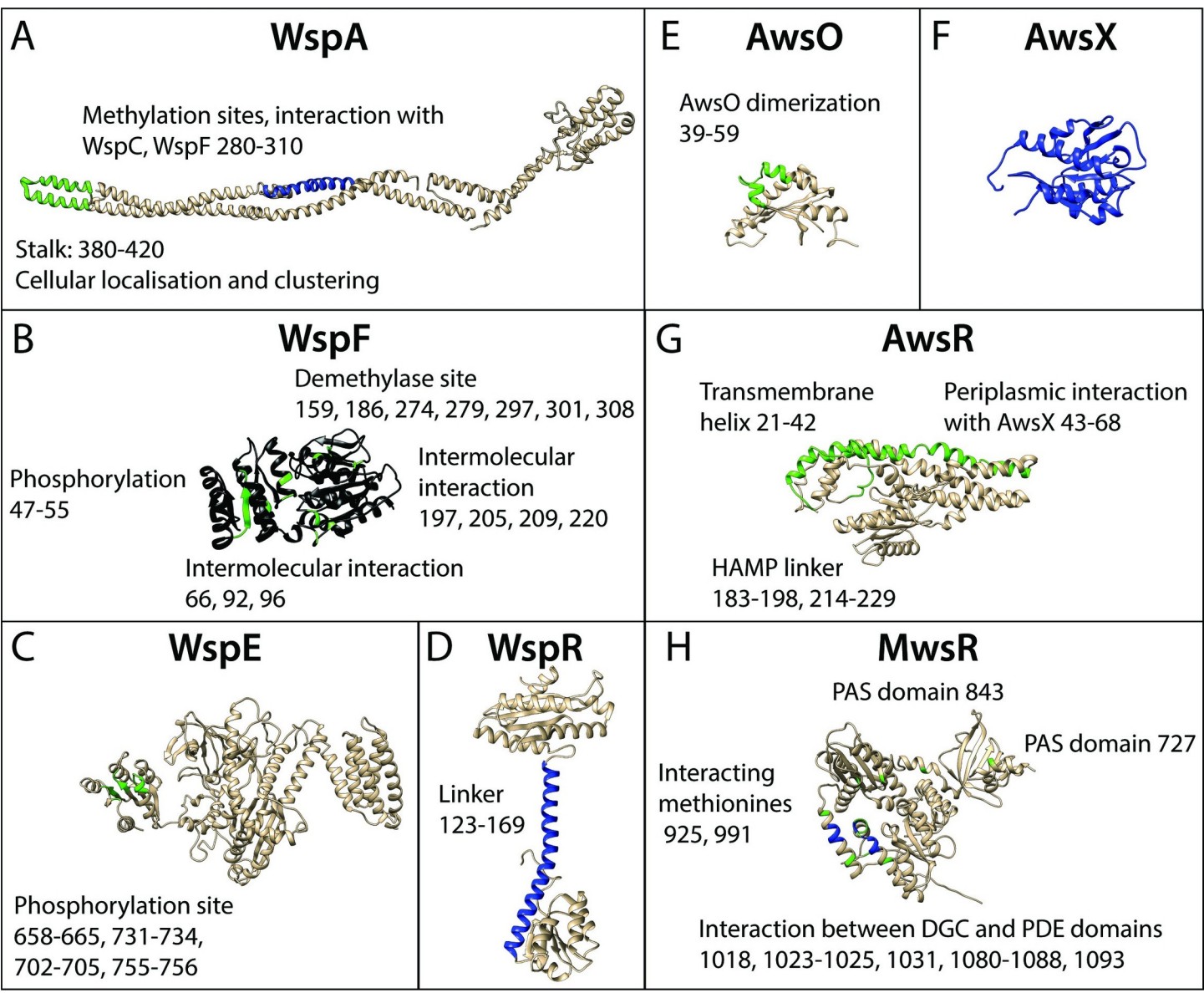

**Fig 2. Predicted mutational targets and proposed molecular effects.** Black represents any inactivating mutation including frame shifts, blue represents in frame inactivating mutations, green represents amino acid substitutions. Numbers refer to amino acid residue numbers in Pf-5. (*A*) WspA–amino acid substitutions are expected at the tip of the stalk and in-frame deletion of methylation sites. (*B*) WspF–any inactivating mutation is predicted, amino acid substitutions are predicted only in areas where they disrupt intermolecular interactions. (*C*) WspE–amino acid substitutions are predicted near the phosphorylation site. (*D*) WspR–small in frame deletion and amino acid substitutions in the linker is predicted to cause constitutive activation. (*E*) AwsO–amino acid substitutions disrupting AwsO dimerization is predicted to lead to increased binding to AwsX without the presence of an activating signal. (*F*) AwsX–any inactivating mutation that keep the reading frame intact and do not interfere with expression of downstream AwsR is predicted. (*G*) AwsR–amino acid substitutions in the periplasmic region or transmembrane helix that disrupt the interaction with AwsX or to the HAMP linker is predicted. (*H*) MwsR–mutations are predicted in the interface between the DGC and phosphodiesterase domains and in the most C-terminal of the PAS domains resulting in constitutive activation.

protein, for example a complete loss of function, will have similar fitness effects [43–47]. Based on the limited data from SBW25 we cannot predict which mutated proteins will produce the most fit mutants.

The outcome of experimental evolution will likely be influenced by the combined effects of mutant fitness and the mutation rates to adaptive phenotypes, which is determined by

mutational target size, i.e. number of sites, and mutation rate heterogeneities among these sites. The development of an improved, more principled, predictive model requires data from more than one strain as it will depend on the relative importance of these factors and to what extent they are conserved between different species or can be predicted *a priori*.

## Experimental test of forecasts in *Pseudomonas protegens* Pf-5

To determine if the WS system can be extended to a related species and to experimentally test evolutionary forecasts in Pf-5, we inoculated 60 independent wells with the wild type for five days of experimental evolution under static conditions. The environmental conditions were changed compared to the SBW25 system to test if a dominant selective pressure, access to oxygen at the air-liquid interface, can be established that is not strictly dependent on the specific experimental conditions. We changed growth medium (KB changed to TSB with glycerol and $MgSO_4$), temperature (28˚C to 36˚C), growth vessel (glass tube to polypropylene deep well plate) and final population size (about $2 \times 10^{10}$ to $4 \times 10^8$). Air-liquid interface colonization was observed for most of the wells and independent mutants with clearly visible changes in colony morphology were isolated from the populations in 43 wells. One colony for each well was selected for further characterization at random based on a pre-determined position on the agar plate. For the remaining 17 wells, no mutants with clear changes in colony morphology were observed, but this does not exclude that such mutants could be present in the population at a lower frequency (<1%) or that other adaptive mutants were present (see S1 Text for extended discussion).

## Most mutations are found in the Wsp, Aws and Mws pathways

We started with evaluating our predictions on the genetic level, in terms of types of mutations, DGC pathways and mutational targets, as it is relatively straightforward to find mutations using DNA sequencing. Identifying the causal mutations also provides clues to the phenotypic basis of air-liquid interface colonization and is needed to be able to reconstruct mutations to determine causality and measure fitness in the absence of possible secondary mutations.

Mutations were identified using a combination of Sanger and Illumina genome sequencing (Fig 3 and S2 Table). As predicted (Prediction 5), the majority (40/43) of mutations were associated with the Wsp, Aws, and Mws DGC pathways (Fig 1D) that are subject to negative regulation (Fig 1C). In addition, the prediction that promoter mutations would be the second most common type of mutation (Fig 1C) was accurate (Prediction 4), with two mutations found upstream of the *aws* operon that were predicted to disrupt the terminator of a high expression ribosomal RNA operon representing a putative promoter capture event. Promoter mutations were also found upstream of PFL_3078, which is the first gene of a putative polysaccharide locus (PFL_3078–3093) that has recently been characterized *in silico* and named *Pseudomonas* acidic polysaccharide (Pap) [48]. Pap is a newly discovered polysaccharide encoded by PSF113_1955–1970 in *Pseudomonas fluorescens* F113 [48], which is only present in closely related strains (and not in SBW25). The operon encodes genes typical and necessary for the biosynthesis of a polysaccharide [48] and it is likely that PFL_3078–3093 encodes the genes for biosynthesis of the main biofilm structural component used by these mutants.

The previously developed mathematical null model ([26], S1 Text) successfully predicted (Prediction 6) that of the three common pathways to WS, Wsp would be the most common (16 mutants) followed by Aws (14) and then Mws (10) (Fig 1E). Mutations were predominately found in the negative regulators WspF (15 mutants) or AwsX (9), but also in interacting proteins WspE (1) and AwsR (3) (Fig 1E). Given that the mutational target size is expected to be smaller for the interacting proteins (Fig 2) this is not surprising. However, no mutations were

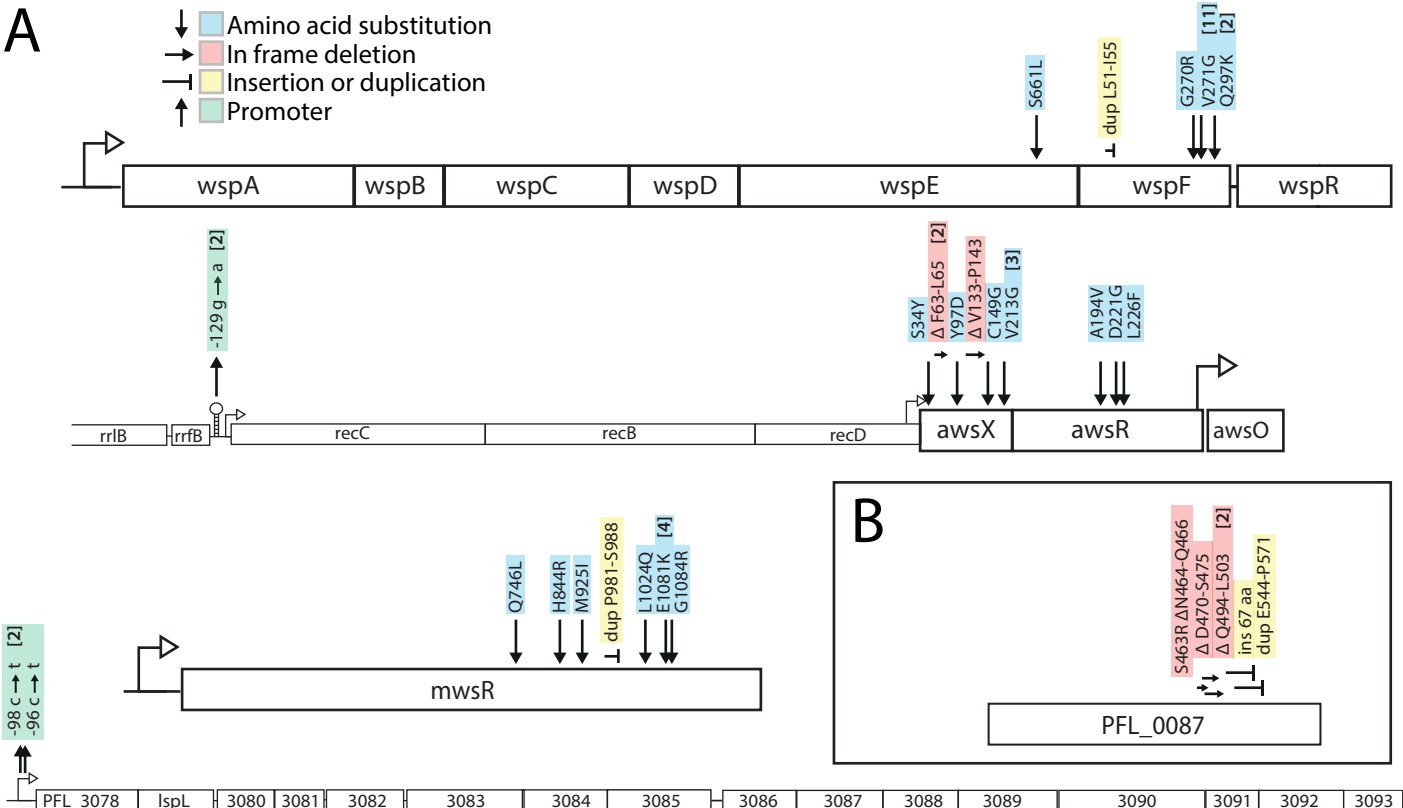

**Fig 3.** (*A*) Forty-three independent mutants of wild type *Pseudomonas protegens* Pf-5 were isolated after experimental evolution based on their divergent colony morphology and mutations were identified in four operons. Numbers in brackets are the number of independent mutants found. Details are available in S2 Table. (*B*) Experimental evolution with a Δ*wsp* Δ*aws* Δ*mws* triple deletion mutant resulted in WS types with mutations in the DGC PFL_0087/DgcH.

found in WspA despite a predicted high rate (Fig 1E). The majority (16 out of 22) of the mutated sites or regions was predicted (Prediction 7) for WspF, WspE, AwsX, AwsR and MwsR (Fig 2), but in most cases they were not identical to those in SBW25 (S2 Table). A further analysis of the likely functional effects of mutations and how they change reaction rates in the mathematical model of the genotype-to-phenotype map is available in the S1 Text and S2 Table.

Identical WspF V271G missense mutations accounted for 11 out of 15 mutations in this protein, which could indicate a putative mutational hot spot site or a greatly increased fitness relative to other WS mutants. The previously described mutational hot spots sites in SBW25 in the *awsX*, *awsR* and *mwsR* genes [26] appeared absent, demonstrating how mutation rate differences can skew evolutionary outcomes even for closely related species with similar genotype-to-phenotype maps. Mutations in WspA were predicted to be one of the major mutational routes to WS based on the mathematical model (Prediction 6, Fig 1E, [26]), but no mutations were found either in this study or in SBW25 [23]. However, when the mutational spectrum of WS mutants in SBW25 was determined in the absence of selection for growth at the air-liquid interface, WspA mutants occurred at rates similar to those of WspE and WspF, as predicted by the model, and their low frequency after experimental evolution could be explained by their lower fitness [26]. To test if a WspA mutation could cause a WS phenotype in *P. protegens* and measure its fitness, a common deletion mutation found in SBW25 (WspA

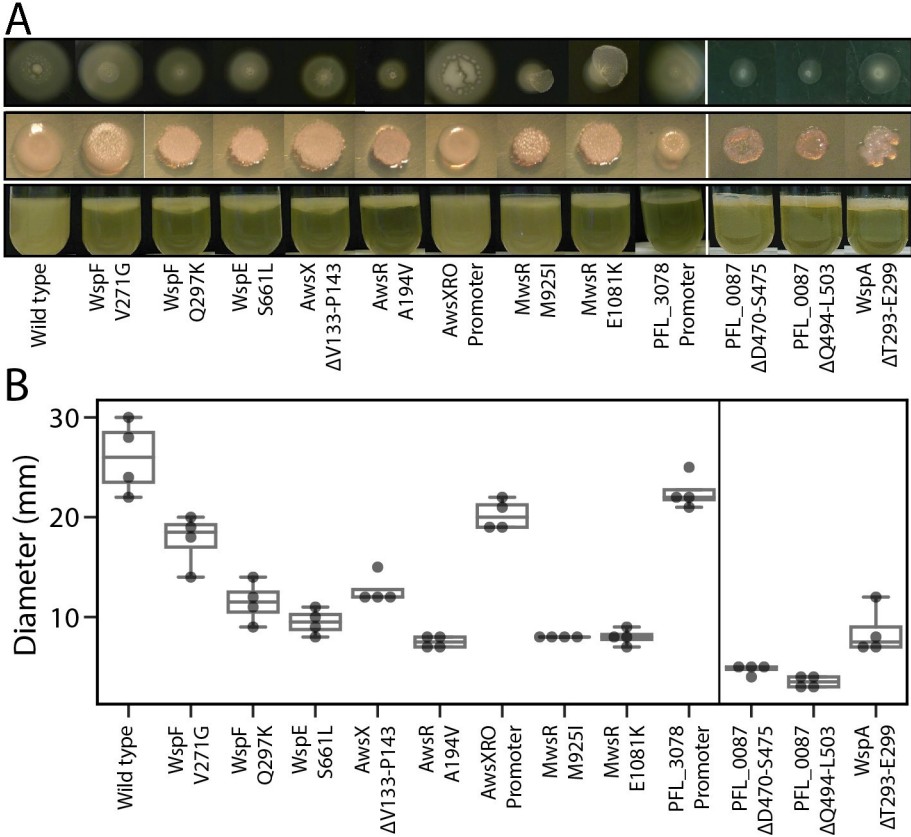

**Fig 4. Phenotypic characterization of reconstructed mutants.** (*A*) Motility, colony morphology and air-liquid interface colonization of reconstructed representative mutations. (*B*) As expected, motility was significantly reduced for all mutants if the c-di-GMP network was activated but was only slightly reduced for the AwsXRO promoter capture. Additionally, motility of the PFL_3078 promoter mutant was not significantly reduced, but this mutant was not expected to have increased c-di-GMP levels (one-way ANOVA, $F_{(12,39)}$ = 63.6, p < 0.0001, pairwise differences assessed with student *t*-tests with α = 0.05). Four replicates on two separate plates per strain were used.

T293-E299) was introduced into the native gene in Pf-5, resulting in a typical wrinkly colony morphology (Fig 4A).

To investigate if there are also rare pathways to the WS phenotype, the entire *wsp*, *aws*, and *mws* operons were deleted and experimental evolution was repeated as previously done for SBW25 [21]. Despite inoculating 60 wells with Pf-5 Δ*wsp* Δ*aws* Δ*mws*, as was done with the wild type, WS types were only detected in 7 wells, which is likely due to a lower mutation rate to WS when the common pathways are absent. Mutations in the DGC PFL_0087/DgcH accounted for six out of the seven WS types found (Fig 3B). This was also the dominant pathway in the SBW25 Δ*wsp* Δ*aws* Δ*mws* strain where mutations in the corresponding region of PFLU0085 were responsible for 47% of WS mutants [21].

## Adaptive mutants colonize the air-liquid interface and most have WS morphology and decreased motility

We evaluated our phenotypic predictions in terms of ability to colonize the air-liquid interface (Fig 1A) (Prediction 1), colony morphology caused by overproduction of EPS (Fig 1B) (Prediction 2), and reduced motility (Fig 1D) (Prediction 3). Due to the difficulties of constructing large numbers of mutants in non-model species, it was necessary here to focus on a select set

of diverse mutants. Representative mutations were reconstructed in the wild type using an allelic exchange protocol to determine that the identified mutations were the sole cause of air-liquid interface colonization and colony morphologies, and to exclude the influence of secondary mutations before further characterization (Fig 4A). Two different phenotypes were observed among the Pf-5 mutants. First, mutants similar to the original WS type in SBW25 had a clear motility defect and mutations in the Wsp, Aws, Mws, and PFL_0087/DgcH pathways (Fig 4A and 4B)). The second phenotype observed was less wrinkly and had similar motility as the wild-type (Fig 4A and 4B)) and mutations upstream of AwsXRO and the PFL_3078–3093 operon. The phenotypic differences of the AwsXRO putative promoter capture compared to other Aws mutants could be due to a lower activation of the c-di-GMP network or by its over-expression being linked to a ribosomal RNA promoter that is highly expressed only under fast growth. That is, as growth rate decreases, the cells are expected to reduce EPS synthesis and increase motility. Mutations in the promoter region of the PFL_3078–3093 operon are not expected to increase c-di-GMP levels, and thus these mutants are not expected to exhibit reduced motility unless expression of the Pap polysaccharide [48] in itself hinders movement.

## Mutants vary in competitive fitness and rapidly invade wild type populations

To evaluate several of our predictions we need to measure the fitness of the mutants under similar conditions as during experimental evolution. This include predictions that the adaptive mutants can invade a wild type population and that the rarity of certain predicted mutants, for example WspA, are due to lower fitness compared to other WS types (Prediction 8). Fitness measurements can also provide important data on if the relative fitness effects of mutations are conserved between species. Two types of fitness assays were performed as previously described [21] to measure differences in fitness. The first assay measures "invasion fitness" where the mutant is allowed to invade a wild type population from an initial frequency of 1%. This confirms that the mutations are adaptive and that mutants can colonize the air-liquid interface. The invasion assays showed that all reconstructed mutants could rapidly invade an ancestral wild type population (Fig 5A and S3 Table). Although there were significant differences between selection coefficients of the mutants (one-way ANOVA, $F_{(12,68)} = 73.7$, $p < 0.0001$, pairwise differences assessed with student $t$-tests with $\alpha = 0.05$), no mutant was significantly different from the most common mutant (WspF V271G, two-tailed $t$-test $p > 0.01$).

The second fitness assay measures "competition fitness". Here, each mutant is mixed 1:1 with the most common WS type (WspF V271G) at the start of the competition [21]. The competition assay also showed that the ancestral wild type was rapidly outcompeted by a WS mutant at a 1:1 initial ratio (Fig 5B and S3 Table). There was significant variation in fitness between the WS mutants (one-way ANOVA, $F_{(12,43)} = 25.5$, $p < 0.0001$, pairwise differences assessed with student $t$-tests with $\alpha = 0.05$). The PFL_3078 promoter mutant had the lowest fitness ($s = -0.1$, two-tailed $t$-test $p < 0.0001$) meaning that it is expected to be rapidly outcompeted by the WS mutants (Fig 5B) (Prediction 2). The PFL_0087 mutants that were only found when the common pathways were deleted had lower fitness (two-tailed $t$-tests $p = 0.0005$, $p = 0.001$) and this was also true for the constructed WspA mutant (two-tailed $t$-test $p = 0.002$), which could explain why these were not found after experimental evolution with the wild type strain (Prediction 8).

## Pel contributes to air-liquid interface colonization together with another unknown adhesion factor

WS mutants form stable mats at the air-liquid interface, but the molecular solution to increased cell-cell and cell-surface adhesion needs to be determined to evaluate our phenotypic

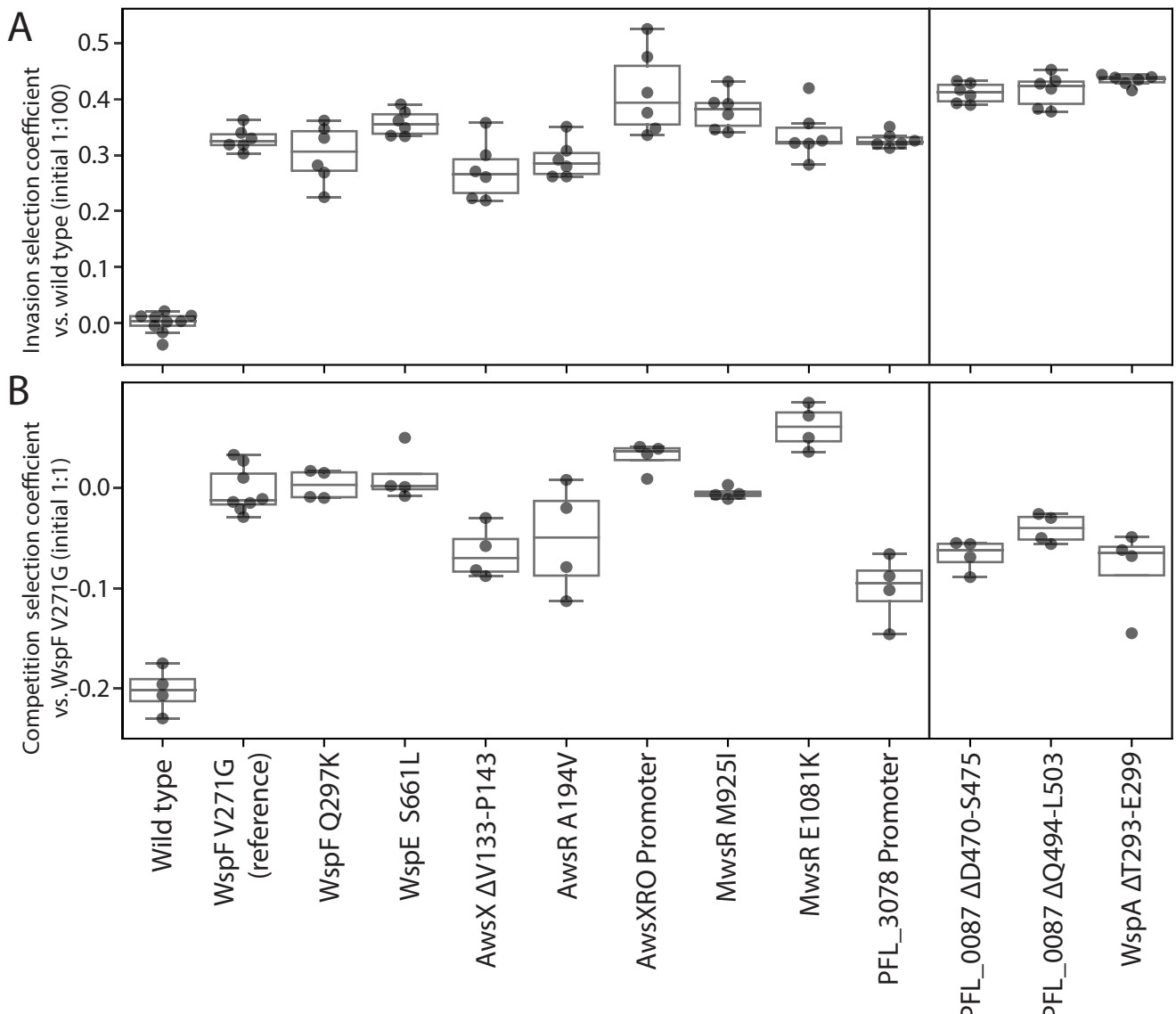

**Fig 5. Fitness of reconstructed *P. protegens* Pf-5 WS mutants was measured in pairwise competitions.** (*A*) Invasion fitness was measured relative a dominant ancestral wild type strain with a 1:100 initial ratio. All mutations were adaptive and can increase from rare to colonize the air-liquid interface. Six independent competitions were performed for each pair. (*B*) Competition fitness was measured relative the most common WS mutant (WspF V271G) in a 1:1 initial ratio to compare the fitness of different WS mutants and the alternative phenotypic solutions. Four independent competitions were performed for each pair.

prediction further. To test if the Pel exopolysaccharide is the main structural component (Fig 1B) used by the different WS mutants of Pf-5 (Prediction 2), the *pelABCDEFG* operon (PFL_2972-PFL_2978) was deleted from Pf-5 and combined with previously characterized WS mutations and fitness was measured. Both invasion fitness (Fig 6A) and competition fitness (Fig 6B) were significantly lower (two-tailed t-tests p < 0.01) compared to isogenic strains with an intact *pel* operon (Fig 5A and 5B, average fitness of mutants with intact *pel* are plotted as red triangles in Fig 6 and S3 Table) except invasion fitness for the AwsX mutant (two-tailed *t*-tests p = 0.08, one outlier). Deletion of the *pel* operon in the absence of WS mutations did not significantly reduce fitness in invasion against the wild type (0.020 +/-0.050 (SD), two-

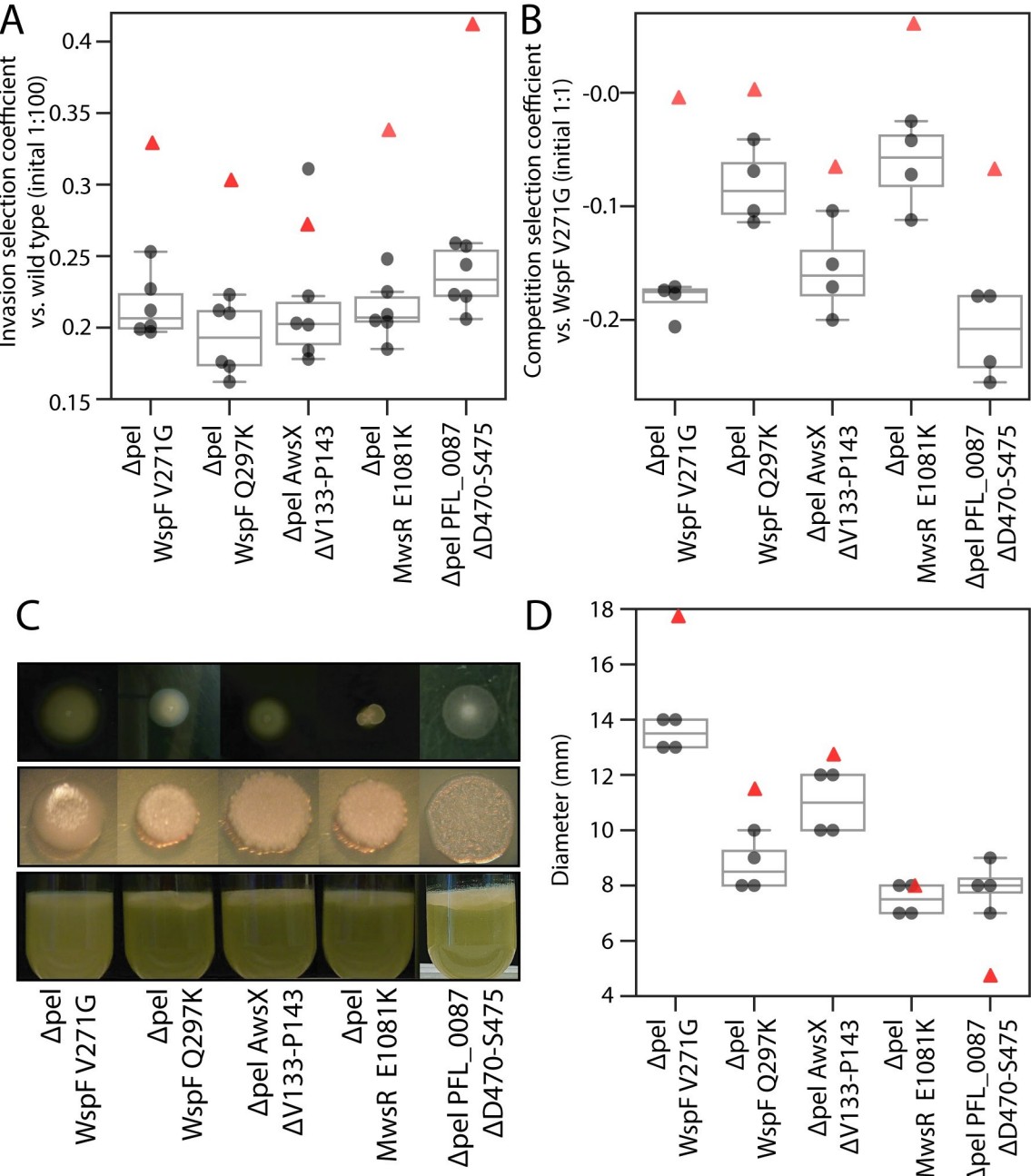

**Fig 6. Contribution of *pel* to WS phenotype and fitness.** (*A*). Deletion of *pel* in WS mutants reduces invasion fitness (mean values of intact *pel* mutants plotted as red triangle in all plots). Fitness of reconstructed *P. protegens* Pf-5 WS mutants without the *pel* operon was measured in pairwise competitions. Invasion fitness was measured relative a dominant ancestral wild type strain with a 1:100 initial ratio. Six independent competitions were performed for each pair. (*B*) Deletion of *pel* in WS mutants reduces competition fitness. Competition fitness was measured relative the most common WS mutant (WspF V271G) in a 1:1 initial ratio. Four independent competitions were performed for each pair. (*C*) Deletion of *pel* in WS mutants did not result in ancestral smooth colony morphology or loss of ability to colonize the air-liquid interface suggesting a secondary EPS component is produced. (*D*). Deletion of *pel* did not restore motility showing that Pel overproduction is not the cause of the motility defect in WS mutants. Four replicates on two separate plates per strain were used.

tailed *t*-test p = 0.38) or in competition against the wild type (-0.011 +/-0.024 (SD) two-tailed *t*-test p = 0.49). This suggests that Pel polysaccharide serves as an important structural component for colonizing the air-liquid interface and that Pel production is activated by mutations leading to increased c-di-GMP levels. Although deletion of the *pel* operon in WS mutants resulted in less wrinkly colony morphology, it did not result in a smooth ancestral type. Neither did deletion of *pel* abolish the ability to colonize the air liquid interface (Fig 6C) or the ability to invade wild type populations (Fig 6A). This suggests that increased c-di-GMP levels induce production of an additional adhesive component, at least in the absence of *pel*. In SBW25, *pgaABCD* encodes proteins for biosynthesis of an alternative c-di-GMP-controlled EPS that is used when the cellulose biosynthesis operon is deleted [22]. Deletion of the *pgaABCD* operon (PFL_0161-PFL_0164) or the Psl biosynthetic locus (*pslABCDEFGHIJKN*, PFL_4208-PFL4219) in Pf-5 strains with deletion of the *pel* operon combined with WS mutations did not result in a reversion to a wild type phenotype. As expected, if the motility defect observed for WS mutants is primarily caused by high c-di-GMP levels rather than high production of Pel, the motility was also reduced for WS mutants with *pel* deleted (Fig 6D, intact *pel* WS mutants plotted as red triangles).

## Predictions for other *Pseudomonas* species

Many of our predictions for Pf-5 (Fig 1) are also applicable to other *Pseudomonas* species and in showing that the WS system can be extended to a related species we lay the foundation for a diverse experimental system for testing evolutionary forecasts. This can provide unbiased tests of our ability to predict short-term evolutionary processes and answer fundamental questions about the conservation of genotype-phenotype-fitness maps and mutational biases needed to make general forecasts. Five other *Pseudomonas* species (Fig 7 legend) were chosen to represent the phylogenetic diversity of *Pseudomonas* (S3 Fig) [49] and considering their complement of DGCs and EPSs (Fig 7A and 7B and S4 Table) [48] to make further predictions. All species are obligate aerobes except for the facultative anaerobes *P. aeruginosa* and *P. stutzeri* that are also expected to grow faster with access to oxygen and respond to selection for colonization of the air-liquid interface. These species encode from none to all three of the main DGCs used in SBW25 and only three species contain genes related to cellulose biosynthesis, the main EPS in SBW25 (Fig 7 and S4 Table). A summary of predictions is shown in Table 1.

The experimental results for Pf-5 largely agrees with predictions and therefore the new data do not give reason for major updates to genetic predictions. The main difference for the other species is that not all commonly used pathways (Wsp, Aws and Mws) are present for all species. While there is some support of conservation of relative fitness effects of mutations in different genes between Pf-5 and SWB25 in that WspF and WspE mutants are fitter than WspA mutations, despite their different EPS components, it remains to be seen if this is also the case over larger phylogenetic distances. The low frequencies of identical mutations and lack of conservation for mutational hot spot sites suggest that, at present, a more detailed prediction of exact mutations is not possible and only mutated regions and functional effects can be predicted as done here, but with incorporation of new mutational data from Pf-5 (Fig 2, see S1 Text for extended discussion).

## Discussion

The ability to forecast evolutionary processes could contribute to addressing major societal problems in key areas including infectious diseases, cancer, climate change and biotechnology [7, 50–52]. For many applications, at least in the near future, evolutionary repeatability alone is likely to be sufficient for making forecasts by utilizing large datasets combined with machine

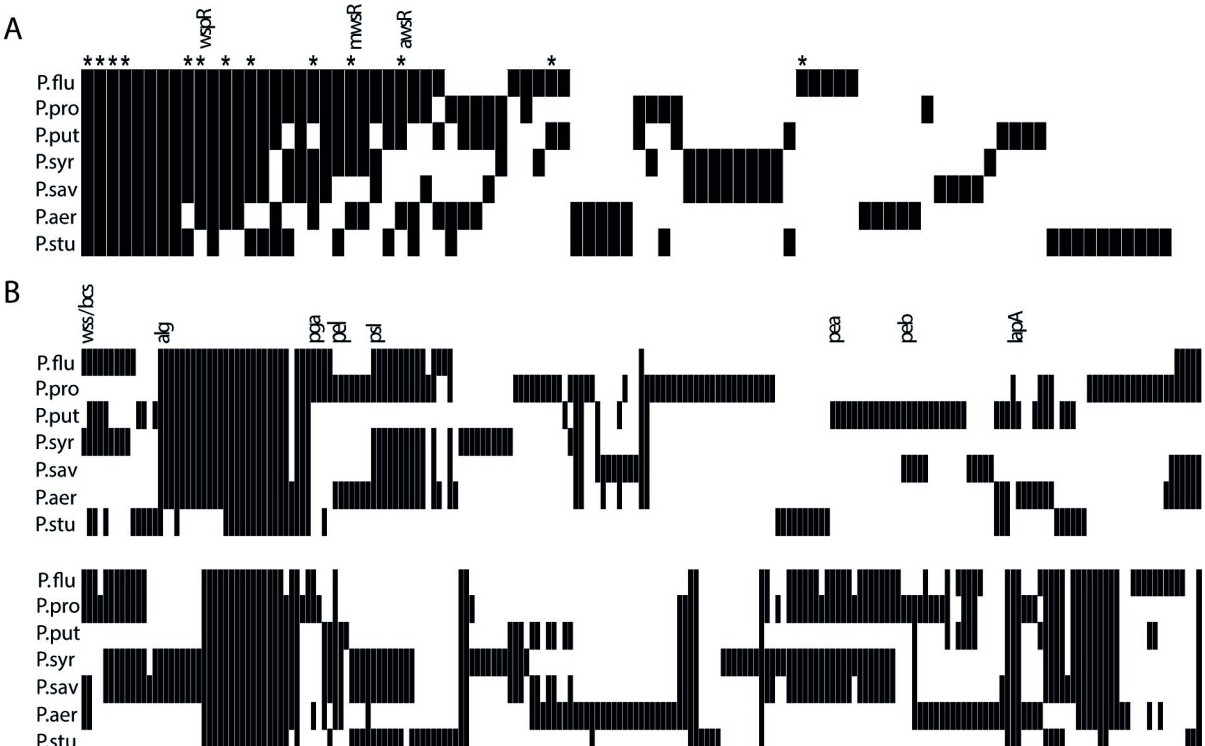

**Fig 7. Diversity of DGCs and biofilm-related genes for seven *Pseudomonas* species.** (*A*) Five other *Pseudomonas* species (*P. putida* KT2440, *P. syringae* pv. tomato DC3000, *P. savastanoi* pv. phaseolicola 1448A, *P. aeruginosa* PAO1, *P. stutzeri* ATCC 17588) were chosen based on phylogenetic diversity to extend predictions. Including *P. fluorescens* SBW25 and P. *protegens* Pf-5, the seven species encode 251 putative DGCs, divided into 87 different homolog groups of which 8 are present in all genomes. WS mutations in SBW25 have been found affecting 13 of these DGCs (marked with *) with an additional nine that have been detected only in combinations with other mutations. SBW25 and Pf-5 share 33 DGCs with 6 unique for each species. It should be noted that not all DGCs are likely to be catalytically active. (*B*) Diversity of biofilm-related genes including putative EPSs, LPS modification, cell chaining, adhesins and known regulators. The genomes of the *P. aeruginosa* strains PA7, UCBPP-PA14 were also included in the analysis and results were in most cases identical to PAO1 (not shown in Fig 7) except for an absence of homologues for EPS genes *pelA-D* and the DGCs PA2771 in UCBPP-PA14 and PA3343 in PA7. Detailed information is available in S4 Table.

learning or statistical models. However, there is a clear advantage of also developing forecasting methods that rely on an understanding of the causes of repeatability because they can be used to test fundamental assumptions about evolutionary processes and provide forecasts for novel situations for which no previous data is available. Such novel situations would also include attempts to control and change evolutionary outcomes by interventions guided by forecasting models. Although there is an extensive theoretical framework for understanding the predictability of evolution, in terms of, for example, fitness landscapes and epistasis [8, 53], predictions about the evolutionary outcome for a specific novel situation are rare and models often use non-accessible parameters or lack explicitly testable hypotheses for the causes underlying failures, such as invoking historical contingency [5].

In this work we explain the basis of our forecasts of experimental evolution, test them experimentally, investigate reasons for failures and extend forecasts to other species to be tested in future studies. We show that this extended wrinkly spreader system has several characteristics that makes it suitable as a model for testing evolutionary forecasts in that the outcomes are not specific to a particular strain or environment, a dominant selective pressure can be established and there are multiple genetic and phenotypic solutions to the adaptive challenge. The selection for increased cell-cell adhesion and biofilm formation is also likely to be

**Table 1. Predictions for other *Pseudomonas* species.**

| Species | Specialization | Primary WS structural component | Alternative WS structural component | Alternative phenotypes | Types of mutations | Order of pathways | Top mutated genes | Genes with promoter mutations | Genes with promoter captures |
|---|---|---|---|---|---|---|---|---|---|
| *P. fluorescens* SBW25 | Air-liquid interface colonization | Cellulose | PGA | LPS, cell-chaining, CPS | 1. Loss-of-function 2. disabling of interaction 3. promoter and promoter capture 4. activating 5. double loss-of-function | Wsp, Aws, Mws, PFLU0085 (DgcH) | *wspF*, *wspE*, *awsX*, *awsR*, *mwsR* | PFLU0956, PFLU5698, PFLU1349, PFLU0621 | PFLU0183, PFLU4306, PFLU4308 |
| *P. protegens* Pf-5 | Air-liquid interface colonization | Pel | multiple or uncharacterized | PFL_3078–3095 | | Wsp, Aws, Mws, PFL_0087 (DgcH) | *wspF*, *wspE*, *awsX*, *awsR*, *mwsR* | PFL_3078–3095 | *awsXRO* |
| *P. putida* KT2440 | Air-liquid interface colonization | Cellulose | Pea, Peb | LPS, cell-chaining, CPS | | Wsp, Aws, Mws, DgcH | *wspA*, *wspF*, *wspE*, *awsX*, *awsR*, *mwsR* | | |
| *P. syringae* DC3000 | Air-liquid interface colonization | Cellulose | | LPS, cell-chaining, CPS | | Wsp, Mws, DgcH | *wspA*, *wspF*, *wspE*, *mwsR* | | |
| *P. savastanoi* 1448A | Air-liquid interface colonization | Psl | uncharacterized | LPS, cell-chaining, CPS | | Wsp, DgcH | *wspA*, *wspF*, *wspE* | | |
| *P. aeruginosa* PAO1, PA14, PA7 | Air-liquid interface colonization | Psl (Pel for PA14) | Pel | LPS, cell-chaining, CPS | | Wsp, Aws, Mws, DgcH | *wspA*, *wspF*, *wspE*, *awsX*, *awsR*, *mwsR* | | |
| *P. stutzeri* ATCC1758 | Air-liquid interface colonization | Cellulose | | LPS, cell-chaining, CPS | | DgcH or unknown under negative regulation | *dgcH* or unknown under negative regulation | | |

relevant ecological traits for natural populations as it can increase antibiotic resistance and immune evasion and decrease protozoan predation [54, 55]. Indeed, clinical isolates with wrinkly/rugose colony morphology are commonly found in *Pseudomonas aeruginosa* strains causing chronic infections in the lungs of people with cystic fibrosis and mutations are found in the same genes as predicted here, including *wspF*, *awsX* (*yfiR*) and *mwsR* (*morA*) [40, 56, 57]. However, there are also substantial limitations for this model system in terms of the short time scale of the experiments and the difficulty of incorporating fitness data into predictive models because fitness is frequency-dependent, so there is a clear need for additional model systems where clear predictions can be made and tested.

Our forecasts were accurate in terms of mutants evolving to colonize the air-liquid interface (Fig 1A, Prediction 1), use of EPS activated by c-di-GMP (Fig 1B, Prediction 2 and 3), types of mutations (Fig 1D, Prediction 4), pathways used (Fig 1E, Prediction 5 and 6), and mutated regions (Fig 2, Prediction 7). These successes are based on the conservation of the fitness of different phenotypes and the conservation of the genotype-to-phenotype map between SBW25 and Pf-5. Determining the reasons for failed forecasts are perhaps more interesting in terms of understanding the limits of our predictive ability and main challenges for moving forward with the development of more principled predictive models, including defining the minimal information needed to make forecasts, what can be predicted *in silico* and what needs to be measured experimentally.

First, strain-specific adaptations pose a major challenge for successful forecasting. Here, the most fit of the two phenotypes used the Pel EPS, which was one of the candidate EPS that could be predicted from previous data, as well as another yet unidentified structural component. However the other phenotypic solution used a recently identified, though

uncharacterized, novel polysaccharide [48] encoded by PFL_3078–3093 for air-liquid interface colonization, demonstrating the difficulty posed by novel strain-specific phenotypic solutions. In contrast to SBW25, where all promoter mutations resulted in up-regulation of DGCs and reduced motility [21], the mutation upstream of PFL3078-3093 demonstrates the possibility of direct transcriptional activation of polysaccharide components that are not under post-translational control of c-di-GMP. It is also clear that strains of the same species can initiate biofilm formation in very different ways, as demonstrated for *Pseudomonas aeruginosa* PAO1 and PA14 [58], and that such differences would be difficult to predict without experimental data, which could potentially have a major impact on the predictability of evolutionary outcomes in an extended WS system.

A second challenge is the prediction of relative fitness for different wrinkly spreader mutants. For the multi-protein pathways Wsp and Aws, the mathematical model (S1 Text, Model IV in [26]) successfully predicted (Fig 1E) the top mutated proteins except in the case of WspA where no mutations were found. WspA mutants were not found in the original study in SBW25 either [23], but this was shown to be due to lower fitness relative to WspF and WspE mutants rather than a lower mutation rate to WS [26]. This is also a likely explanation for the absence of WspA mutants here (Fig 5B), but it remains to be determined if this fitness difference is conserved in other species or if sometimes WspA mutants are more fit. Prediction of the relative fitness effects of adaptive mutants remains a daunting challenge, but it is interesting to note that for both Pf-5 and SBW25 high fitness WS types have mutations in the same proteins (WspF, WspE, MwsR) and low fitness WS mutations were found in WspA, AwsX, AwsR, and PFLU0085/PFL_0087 for both species. If this trend of conservation would extend to other species as well, it could increase the potential of forecasting by partially solving the fundamental problem of how to predict the relative fitness effects of mutations. A possible complication is that relative fitness might be influenced to a great extent by the environment, including the frequency of other adaptive mutants in the population. An alternative way forward might be to develop metabolic fitness models [59] to predict the trade-off between the metabolic cost of producing exopolysaccharide and global effects on gene regulation from increased levels of c-di-GMP with the benefit derived from increased ability to colonize the air-liquid interface. But it is likely that such models would require extensive experimental data and if new data is required for each species or for every new environment, more success might be achieved by measuring the fitness effect of a few targeted mutations in predicted genes to identify those with highest fitness (Prediction 8).

The third major challenge is predicting differences in mutation rates as mutational hot spot sites do not appear to be conserved between SBW25 and Pf-5. Although a mathematical null model that incorporates information about the Wsp, Aws, and Mws molecular networks (Fig 1E and S1 Text, Null model IV in [26]) successfully predicted that Wsp would be the most commonly used pathway to WS (predicted 54%, observed 40%) followed by Aws (predicted 30%, observed 35%), and Mws the most rare (predicted 16%, observed 25%), the high frequency of Wsp mutants is mainly due to repeated V271G mutants. The repeated isolation of this mutant seems to be caused by a putative mutational hot spot site, given that it does not appear to have higher fitness than other WS mutants (Fig 5), but it is possible that its fitness could be higher under the specific conditions where it first evolved. It is also worth noting that direct use of mutation rate data from SBW25 [26] would result in poorer predictions than the mathematical null model due to mutational hot spot sites in *awsX*, *awsR* and *mwsR* for that species, demonstrating the usefulness of constructing general mechanistic models rather than relying solely on experimental data.

It is possible that an increased mutation rate at promoters [60] is also responsible for the relatively high frequency of mutations upstream of PFL_3078–3093 despite a small mutational

target size and the observation that this mutant is rapidly outcompeted by other WS types (Fig 5B). In SBW25, low fitness phenotypes that colonize the air-liquid interface based on LPS modification or cell-chaining, are observed prior to the rise of WS to high frequencies [22]. This is due to the presence of apparent mutational hot spot sites in these genes, which allow these mutants to appear early during the growth phase despite their relatively small mutational targets [20, 22, 61]. The extent to which mutational hot spot sites contribute to parallel evolution in experimental evolution studies is unclear. This is partly due to a focus on gene level parallelism and lack of information about exact mutations in databases [62], but the primary problem is the inability to determine the reasons for parallelism in terms of increased mutation rate, differential fitness effects of mutations, and mutational target size [63]. In *E. coli*, the most common cases of exact nucleotide parallelism described is associated with IS elements that mediate deletions and duplications through homologous recombination or insertions into genes, while parallelism for point mutations was less common [6, 63]. In addition to differences in rates between different base pair substitutions, increased mutation rates associated with nucleotide repeats have also been reported [64, 65]. However, general knowledge of such mechanisms does not allow us to *a priori* predict the distribution of mutation rates for a gene or genome and in many cases there is no obvious mechanistic reason for an observed mutational hot spot site. This difficulty can here be exemplified by attempting to predict hot spot sites in *awsX*, where an in-frame deletion between 6-bp direct repeats account for about 28% of the total mutation rate to WS in SBW25 [26], despite hundreds of possible WS mutations [21, 23, 26, 38], while a 10-bp direct repeat in the same gene is used at a ten-fold lower rate. In Pf-5, only 5 bp of the 6-bp hot spot site in SBW25 is conserved, which could explain why no identical mutations were observed. But another 8-bp direct repeat, where a deletion would cause an in-frame deletion, would be predicted to be used in Pf-5 at a high rate; yet no such mutations are observed among the nine *awsX* mutants found here. It is clear that our current inability to predict mutational hot spot sites can severely limit predictions across species, even though they can make evolution more deterministic in each single case.

In partially predicting evolutionary outcomes in *P. protegens*, this work lays the foundation for future tests of evolutionary forecasts in related *Pseudomonas* species by clearly stating predictions on several different biological levels—from phenotypes to specific regions of proteins that are likely to be mutated. Currently, given what is already known about the effects of unpredictable mutational biases and differences in fitness between different WS types, many of these forecasts will inevitably fail. However, hopefully they will fail in interesting ways, thereby revealing erroneous assumptions and providing possibilities for iterative improvement and development of more principled forecasting models. The ability to remove common genetic and phenotypic pathways provides a unique opportunity to also find those pathways that evolution does not commonly use. This is necessary to determine why forecasts fail and update the predictive models for another cycle of prediction, experimental evolution, and mutant characterization that make it possible to use this iterative model to define the information necessary to predict short-term evolutionary processes.

## Materials and methods

### Strains and media

*Pseudomonas protegens* Pf-5 (GenBank: CP000076.1 [66], previously known as *P. fluorescens* Pf-5) and derivatives thereof were used for all experimental evolution and phenotypic characterization. *E. coli* DH5α was used for cloning PCR fragments for genetic engineering. *P. protegens* Pf-5 was grown in tryptic soy broth (TSB) (Tryptone 17 g, Soytone 3 g, Glucose 2.5 g, NaCl 5 g, $K_2HPO_4$ 2.5 g per liter) supplemented with 10 mM $MgSO_4$ and 0.2% glycerol

(TSBGM) for experimental evolution and fitness assays. For comparison, previous studies of *P. fluorescens* SBW25 used King's medium B (KB) for experimental evolution (20 g proteose peptone #3, 1.5 g $K_2HPO_4$, 1.5 g $MgSO_4\bullet7H_2O$ and 10 ml glycerol per liter) [21–23, 25, 26, 28]. Lysogeny broth (LB) was used during genetic engineering and LB without NaCl and supplemented with 8% sucrose was used for counter-selection of *sacB* marker. Solid media were 1.5% agar added to LB or TSB supplemented with 10 mM $MgSO_4$, 0.2% glycerol and 10 mg/l Congo red. Motility assays were conducted in 0.3% agar TSB supplemented with 10 mM $MgSO_4$, 0.2% glycerol. Kanamycin was used at 50 mg/l for *E. coli* or 80 mg/l for *P. protegens* and gentamicin at 10 mg/l for *E. coli* or 15 mg/L for *P. protegens*. 100 mg/L nitrofurantoin was used to inhibit growth of *E. coli* donor cells after conjugation. All strains were stored at -80˚C in LB with 10% DMSO.

## Experimental evolution

Thirty central wells of a deep well plate (polypropylene, 1.1 mL, round walls, Axygen Corning Life Sciences) were inoculated with approximately $10^3$ cells each from independent overnight cultures and incubated at 36˚C for 5 days without shaking on two different occasions. The wells at the edges of the plate were not used to reduce possible edge effects with increased evaporation and they instead served as contamination controls. A 1 μl plastic loop was used to collect cells from the bottom and the edges and surface of the air-liquid interface that were then transferred to an Eppendorf tube with LB and vortexed vigorously. This sampling means that the frequency of mutants on agar plates is not necessarily representative for the entire population. Suitable dilutions were plated on TSBGM plates with Congo red after 5 days and incubated at 36˚C for 48 h. Plates were screened for colonies with a visible difference in colony morphology and one divergent colony per well was randomly selected based only on its position on the agar plate. In total 43 independent mutants were streaked for single cells twice before overnight growth in LB and freezing. An identical protocol was used for the Δ*wsp* Δ*aws* Δ*mws* strain.

## DNA sequencing

Sanger sequencing was initially used to sequence the *wspF* and *awsX* genes. Seven mutant strains that did not contain mutations in the *wspF* and *awsX* genes were analyzed by genome resequencing. The strains had mutations in *awsR*, *mwsR*, *wspE*, upstream PFL_3078 and in the intergenic region between *rrfB* and *recC* upstream of the *awsXRO* operon. Sanger sequencing of these genes in phenotypically similar mutants was used to find the remaining mutations. Genomic DNA was isolated with Genomic DNA Purification Kit (Thermo Fisher). Sequencing libraries were prepared from 1μg DNA using the TruSeq PCRfree DNA sample preparation kit (cat# FC- 121-3001/3002, Illumina Inc.) targeting an insert size of 350bp. The library preparation was performed according to the manufacturers' instructions (guide#15036187). Sequencing was performed with MiSeq (Illumina Inc.) paired-end 300bp read length and v3 sequencing chemistry. Sequencing was performed by the SNP&SEQ Technology Platform in Uppsala. The facility is part of the National Genomics Infrastructure (NGI) Sweden and Science for Life Laboratory. The SNP&SEQ Platform is also supported by the Swedish Research Council and the Knut and Alice Wallenberg Foundation. Sequencing data were analyzed with using Geneious v. 10.2.3 with reads assembled against the *P. protegens* Pf-5 genome sequence (CP000076.1). Complete sequencing data for all of these clones is available under NCBI BioProject PRJNA737653. Sanger sequencing was performed by GATC biotech and used to sequence candidate genes to find adaptive mutations and to confirm reconstructed mutations. Primer sequences are available in S5 Table.

## Reconstruction of mutations

Thirteen mutations representing all candidate genes found as well as PFL_0087 and WspA mutations were reconstructed in the wild type ancestral *P. protegens* Pf-5 to show that they are the cause of the adaptive phenotype and to assay their fitness effects without the risk of secondary mutations that might have occurred during experimental evolution. A two-step allelic replacement protocol was used to transfer the mutation or deletion constructs into the ancestor as previously described [67], but using the mobilizable pK18mobsac suicide plasmid (FJ437239) (full details are available in the S1 Text).

## Fitness assays

Two types of competition fitness assays were performed similarly to previously described assays [21]. The first assay measures invasion fitness under 48 h, where a mutant is mixed 1:100 with the wild type ancestor tagged with GFP, mimicking early stages of air-liquid interface colonization where a rare mutant establishes and grows at the surface with no competition from other mutants. The second assay measures competition fitness under 24 h in a 1:1 competition against a reference mutant strain tagged with GFP. We chose the WspF V271G mutant as a reference because it was the most frequently found mutant after experimental evolution. Selection coefficients (s) were calculated as previously described (67) as the change in logarithmic ratio over time according to $s = [\ln(R(t)/R(0))]/[t]$, where R is the ratio of mutant to reference and t is the number of generations of the entire population during the experiment. This means that s = 0 is equal fitness, positive is increased fitness, and negative is decreased fitness relative to the reference strain. A full description of the experimental protocol is available in the S1 Text.

## Motility assays

Swimming motility assays were performed in TSBGM plates with 0.3% agar (BD) and the diameter was measured after 24 h of growth at room temperature. Assaying at room temperature, instead at 36°C, reduced in the influence of growth and evaporation to increase reproducibility. Each strain was assayed in duplicates on two different plates, yielding four replicates.

## Bioinformatics analysis of DGCs and EPS genes

Homologs for all DGCs in *P. fluorescens* SBW25 were found using the *Pseudomonas* Ortholog Database (v. 17.2) at Pseudomonas.com [68]. Blast-p searches for GGDEF domains were performed to find remaining DGCs in the six *Pseudomonas* species and their homologs again found using the *Pseudomonas* Ortholog Database [69] and manually inspected. Annotations (Pseudomonas.com. DB version 17.2) were also searched for diguanylate cyclase and GGDEF. Not all DGCs found are likely to have diguanylate cyclase activity, but given the difficulties of predicting which of the partly degenerate active sites are likely to be inactive combined with the possibilities of mutational activation during experimental evolution, none were excluded. There is no simple way to find all genes that can function as structural or regulatory genes to allow colonization of the air-liquid interface. Thus, the selection in Fig 7A and S4 Table should not be considered complete. Putative EPS genes were found using blast-p searches with sequences of known proteins for exopolysaccharide biosynthesis including cellulose, PGA, Pel, Psl, Pea, Peb, alginate and levan. Homologs were found using the *Pseudomonas* Ortholog Database [69] at Pseudomonas.com [68]. Annotations (Pseudomonas.com. DB version 17.2) were also searched for glycosyltransferase, glycosyl transferase, flippase, polysaccharide, lipopolysaccharide, polymerase, biofilm, pili, curli, adhesin and adhesion. Based on previous work in SBW25 and literature searches a few additional genes were added [20–22, 70–73].

## Supporting information

**S1 Table. Sequence identity.**
(XLSX)

**S2 Table. Mutations.**
(XLSX)

**S3 Table. Fitness.**
(XLSX)

**S4 Table. DGCs and biofilm genes.**
(XLSX)

**S5 Table. Primers.**
(XLSX)

**S6 Table. Accession numbers.**
(XLSX)

**S7 Table. Motility data.**
(XLSX)

**S1 Text. Supplementary text and methods, S1-S3 Figs.**
(DOCX)

## Author Contributions

**Conceptualization:** Peter A. Lind.

**Formal analysis:** Jennifer T. Pentz, Peter A. Lind.

**Funding acquisition:** Peter A. Lind.

**Investigation:** Jennifer T. Pentz, Peter A. Lind.

**Methodology:** Jennifer T. Pentz, Peter A. Lind.

**Resources:** Peter A. Lind.

**Supervision:** Peter A. Lind.

**Visualization:** Jennifer T. Pentz, Peter A. Lind.

**Writing – original draft:** Jennifer T. Pentz, Peter A. Lind.

**Writing – review & editing:** Jennifer T. Pentz, Peter A. Lind.

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
