## [Decision Letter · Decision Letter 0]

23 Apr 2021

Dear Dr Lind,

Thank you very much for submitting your Research Article entitled 'Forecasting of phenotypic and genetic outcomes of experimental evolution in Pseudomonas protegens' to PLOS Genetics.

The manuscript was fully evaluated at the editorial level and by independent peer reviewers. The reviewers appreciated the attention to an important problem, but raised some substantial concerns about the current manuscript. Based on the reviews, we will not be able to accept this version of the manuscript, but we would be willing to review a much-revised version. We cannot, of course, promise publication at that time.

If you decide to revise the manuscript for further consideration at PLOS Genetics, please aim to resubmit within the next 60 days, unless it will take extra time to address the concerns of the reviewers, in which case we would appreciate an expected resubmission date by email to plosgenetics@plos.org.

[LINK]

We are sorry that we cannot be more positive about your manuscript at this stage. Please do not hesitate to contact us if you have any concerns or questions.

Yours sincerely,

Olivier Tenaillon

Guest Editor

PLOS Genetics

Kirsten Bomblies

Section Editor: Evolution

PLOS Genetics

Editor's comments to Authors

Dear Jennifer and Peter,

Three independent reviewers have examined thoroughly your work and came up with important considerations that will have to be taken into account for the manuscript to be taken into account.

-The question tackled is complex and the reductionist approach used is relevant but need to be precisely defined and its powers and limits presented.

-My understanding is that a real pedagogic effort was made to explain all the different levels of predictability, unfortunately I agree with reviewer 1 that it makes the paper way too long and finish having the opposite effect of making results less clear. This is not a review paper.

-all reviewers noticed the missing lineages and those should be clearly discussed.

-Reviewer 3 has a strong point when he also mention the change of media. It could be turned into a benefice by noticing that despite these difference similarities were observed but it definitely needs to be addressed.

Reviewer's Responses to Questions

**Comments to the Authors:**

Reviewer #1: This manuscript addresses the predictability of short-term evolution across two species of Pseudomonas, one of which represents an iconic example of diversifying selection in a static environment with previously well described phenotypes and mutational hotspots. The authors ask here if similar phenotypes and similar mutations will evolve in another Pseudomonas species with a considerable divergence from the SBW25 strain previously studied. They also use different media and temperatures to test if the predictions from the previous workout strain adaptation were insensitive to these differences in the environment, as the dominant pressure should be access to oxygen. Furthermore, the authors also performed experimental evolution on selected deletion mutants of the most common pathways in the new species.

The manuscript brings new results on the short-term adaptation to this selective pressure by Pseudomonas Pf5, with a convincing set of well design experiments and an interesting choice of the new strain. It also provides a table of predictions on pathways that are expected to mutate in other species of pseudomonas when under selection to colonize the liquid air interface.

However, I did find that the manuscript was sometimes unclear regarding what the new and the past results indeed were.

In general, I found the manuscript length a bit too long with many repetitions that appear unnecessary. Specially, when the repetitions refer to either hypothesis stated in other papers or models done in previous literature. Thus, the manuscript as currently written does not make rapidly clear to the reader what the novelty here really is.

For example, lines 126 to 143 are a bit repetitive with the introduction and do not appear to me as results. The same applies to lines 156-160.

To me the results start on page 9 Line 190, where phenotypic predictions are made.

The results of the genetic predictions could also be shortened, particularly lines 235- 259, and more directly go to the predictions from the modelling of reaction rates with a clear explanation on what is novel and what comes from previous work by the author(s).

Some minor specific comments are:

Line 287- What is the rational for assuming a 10-fold difference here?

Line 314 – What is meant by “Mutational hotspots with greatly increased mechanistic mutation rates”?

Lines 418-421 It is not clear if this is a mutational hotspot due to high mutation rates of due to selection.

Discussion- The authors should remind the readers that these are forecasts for very short term evolution experiments.

Reviewer #2: The authors present the results of an attempt to forecast phenotypic and genetic changes during a short-term adaptive radiation in static cultures of the bacterial genus Pseudomonas. The present study builds on extensive past work on the evolution of mutants of a particular strain of P. fluorescence colonizing the air-liquid interface (called wrinkly spreaders, WS), and use information about known target genes and phenotypic and fitness effects of mutations in this species to predict mutation targets and their frequencies in a strain of P. protegens after selection under related conditions. Their predictions are largely supported in this new species and under somewhat different conditions, while they also describe failures for some specific predictions, including the role of a novel polysaccharide to form WS mutants and the lack of contribution of mutations in WspA. They close by presenting predictions about the adaptive role of possible genes and pathways to yield WS mutants in a number of other Pseudomonas species based on genome content, but these are left untested.

Aspects we like about this study are the promising role of their experimental model for evolutionary forecasting events of adaptive radiation in other systems, and the careful consideration of the various aspects of the genotype-phenotype-fitness map that are relevant for their predictions, including a critical discussion of possible reasons for the failed predictions. Also, the experiments and analyses of evolved changes, including the construction and analyses of various mutants, are all sound and support the claims. What we feel is lacking, is a more fundamental, mechanistic model allowing such predictions, as many predictions now are direct extrapolations from previous observations in P. fluorescence, rather than predictions based on some underlying level of understanding of the effects and rates of the mutations involved. They do use a (previously published) metabolic model to predict relative contributions of the various WS mutations, and they do mention most likely mutation regions within target genes, and we consider these aspects as most valuable, as they indicate a more fundamental and generalizable way forward. Nevertheless, we think the study is suitable for publication in PLoS Genetics, as it provides a reasonable heuristic attempt to an ambitious goal using an interesting system for evolutionary forecasting of adaptive radiation events. We hope that our comments may help to improve readability and the reader’s appreciation of the implications of their work for the problem of evolutionary forecasting more generally.

Specific comments:

1. It should be stated more clearly for which more general evolutionary questions the WS system is most suitable and promising in the context of evolutionary forecasting. Now, the findings presented are of interest to a rather narrow public of people working with the same system, while this model has clear potential to be relevant more broadly to questions relating to adaptive radiation, biofilm formation, etc. This is now mentioned only briefly at the end, but should be mentioned already in the introduction and discussed more thoroughly in the discussion.

2. The general appeal of the study would also benefit from a better presentation how the successful and failed predictions could be used to develop this system into a more principled predictive model, allowing predictions for other bacterial strains and conditions. For example, would it be possible to link the WS-producing networks of Fig. S2 to basic metabolism, e.g. along the lines recently done for streptomycin resistance in E. coli by Pinheiro et al. (2021, Nat Ecol Evol)? Also, the authors mention (lines 714-721) that mutation hotspots cannot be predicted from genome sequences, while they mention the known roles of repeat sequences and IS-elements. Could that information not be used to a first approximation to determine mutation target size and rates?

3. To aid readers not familiar with the WS system, a schematic overview of the genes and pathways involved in the biofilm formation at the surface of static growth tubes would be valuable addition. Maybe their current figure 1 could be incorporated in such an overview figure. Also, the purpose of Fig. 1 was not fully clear, particularly what prior information was used for the predictions, and that the outcome column summarizes the results of the present study rather than the success of previous predictions with the SBW25 strain. The predictions now seem to be based only on previous empirical observations in SBW25 and not to involve any fundamental reasoning about underlying causes and constraints.

4. 17 out of the 60 Pf-5 lines did not colonize the air-liquid interface. How does this compare to SBW25 and what implication does this have for forecasting the outcome? Might this indicate that this biofilm formation is more constrained in Pf-5? Do these 17 lines represent alternative -rare but more fit- strategies to deal with oxygen depletion?

5. Line 217-219: Unclear why Pel is expected to be the primary EPS in Pf-5, and not one of the other two EPSs mentioned. It should be explained why this EPS is predicted to be preferred over CPS, LPS and PG.

6. Line 303 and further: information about the nature of the different mutations, i.e. whether inactivating or more subtly altering the gene, would be helpful for evaluating the relative mutational targets. For example, how do the authors explain the remarkable repeatability of missense mutation V271G in WspF (line 418), and the absence of nonsense or frameshift mutations? What is their evidence that this is a mutational hotspot, as the authors believe? Is this not a gene that is inactivated by adaptive mutations? Perhaps predictions of these mutation target sizes, at least for SNPs, could be added to Fig. 2?

7. Information about the total number of mutations found in the 43 evolved clones is lacking. The text mentions “secondary mutations”, but not how many. This may also clarify why mutants with single mutations were constructed and analyzed.

8. Line 456: what was tested in the ANOVA presented? It seems to support the statement that the second phenotype was less wrinkly and had similar motility, but unclear how.

9. The section “Predictions for other Pseudomonas species” (p25-27) I found rather superfluous without presenting tests of those predictions. I would prefer a synthesis section summarizing (perhaps with a figure) the possible elements of an improved predictive model of WS evolution, but this might also be done in the discussion (see my 1st comment).

Reviewer #3: In this manuscript, the authors set out to determine if they could expand upon a well-characterized experimental system to predict the phenotypic and genotypic evolution of closely-related bacterial species. This manuscript addresses an important question in evolutionary biology – to what extent is evolution repeatable and thus predictable? Few studies have investigated, at a molecular level, how closely-related bacterial species adapt to a common selective pressure.

The authors find that the genetic routes of adaptation are mostly conserved between two species of Pseudomonas (despite differences in evolution conditions) and identify several routes that are specific to Pf-5 or SBW25. Despite the absence of the primary structural component of SBW25 mats (cellulose), Pf-5 adapts via mutations in the same c-di-GMP/DGC regulatory pathways. The predictions are extended to additional Pseudomonas species, which the authors intend to test in a future study.

MAJOR CONCERNS:

I have several major concerns with the study. First, the study introduced two major variables, genotype and environment, yet the effect of each individual variable on evolutionary outcomes was not thoroughly investigated. The authors assumed that since the same genes were primary targets of selection in both the classical system and the extended system, that any differences in evolutionary outcomes are attributed to genotype rather of environment. I disagree with this assumption.

The authors’ claim of a dominant selective pressure suggests that the shape of the tail of the DFE (and the beneficial mutations that comprise it) remains intact despite the changes between the classical and extended systems. However, due to the substantial environmental changes, there must be a point at which the DFEs diverge, and it is unclear if this point was reached. I would need to see evidence that SBW25 adapts via the same genetic mechanisms in both the classical and extended system (especially beyond Wsp/Mws/Aws) before I’m convinced that other environmental components did not influence adaptation.

Further, I think it is dangerous to rely on the assumption of a single selective pressure when extending predictions to additional species because any divergence in adaptation would be attributed to genetic background. I would be particularly concerned about species that are already well-adapted to colonize the air-liquid interface – thus shifting selection to other environmental components.

A second major concern is that, apart from gene targets, the predictions are too subjective. The predictions should be data-driven, yet some predictions are made without any connection to a data set. For example, there are no citations and no references to data to support the prediction that “mutants that specialize in colonizing the air-liquid interface will have the highest fitness” (lines 190-201). At times, the authors seem to pick and choose what data are used to generate predictions (e.g. that Pel with be the primary EPS based on its importance for pellicle formation at the air-liquid interface in PA 14, lines 218-219). I feel strongly that the authors need to specify the exact data set used to make their predictions.

Finally, I believe that the manuscript would be greatly improved with heavy revisions. Specifically, much of the text lacks clarity. At times, the writing is too vague to convey meaning (e.g. “different results” line 81) and other times where descriptions need to be more concise (e.g. all predictions in the Results). As currently written, the reader needs to expel too much time/effort to follow the paper’s progression and authors’ logic.

MINOR CONCERNS

• It is unclear from the abstract if the focus of the study is the impact of the environment or genotype (or both).

• In the Introduction (line 80), the authors state that “adaptive mutations are often strain specific” but the cited papers do not support this claim.

• The authors did not explain the necessity of the environmental changes. While I presume that some changes were necessary for growth of Pf-5, the abstract states the change was made “to test the robustness of forecasts to environmental variation”. The authors should explain why each of the environmental components were changed in the extended system.

• The authors analyzed 43 clones isolated from 60 total populations. Why weren’t clones analyzed from the other 17 populations?

• Were non-divergent phenotypes considered as adaptive? How can mutants that colonize the air-liquid interface be most fit if no other mutants were analyzed?

o A minor point, but is it fair to say that the observed mutants are most fit since their fitness is density dependent? At high frequencies, they are likely less fit than other competitors.

• This paper that compares the strategies of P. aeruginosa strains PAO1 and PA14 needs to be included in the discussion (DOI: 10.1128/mBio.02644-19)

• Are the entire evolution cultures plated to recover divergent phenotypes or just the mat? Same for the fitness assays. How is the mat disintegrated?

• It is my understanding that the mutants described here were chosen because of their divergent phenotype. There is no data presented in the manuscript that divergent phenotypes are more fit (or better colonizers of the air-liquid interface) than non-divergent phenotypes. The predictions in Fig 1A and Fig 1B were never actually tested. At the very least, the proportion of divergent colonies observed in each population should be included.

• It’s unclear if the whole paper about access to the WS phenotype, divergent phenotypes, colonizers of the air-liquid interface, or whatever mutants are most fit.

• There is no explanation for the claims of “mutational hotspots”. What evidence exists that these loci experience an increased mutation rate? Have you ruled out the possibility that the mutations observed >1x were present in the inoculum – especially WspF V271G?

• The model predicts an abundance of mutations in the Wsp pathway and this was observed. Most of the Wsp mutations were identical (WspF V271G), which the authors infer to be a mutational hotspot. However, the authors state that their model doesn’t take hotspots into account (line 284), so this outcome doesn’t actually support the model.

• Since hotspots were not taken into account in the model, wouldn’t the presence of hot spots skew the

• Throughout the paper, wording needs to be changed to reflect that polysaccharides aren’t encoded by genes (their synthases are).

• I fundamentally disagree with the message in lines 154-168 that evolutionary predictions need to follow a specific progression. I would argue that genotypic and phenotypic predictions can be made independent of one another. Regardless, I don’t think this paragraph needs to exist in the paper. It’s difficult to follow the progression of the paper because the predictions and results follow different order.

• I don’t think the role of c-di-GMP and DGCs (and their impact on matrix formation) are explained well enough for a general audience.

• Were there any batch effects attributed to the two independent evolution experiments?

• The methodology used to predict the location of mutations was not explained (lines 316-318).

• The experiment is not designed such that the most prevalent mutant is necessarily the most fit. Early mutations that allow occupation of the A-L interface have a distinct advantage over late occurring mutations. Therefore, the prevlance of a mutant may be more of a consequence of when the mutation occurred than the magnitude of its fitness. Pathways with a large target size (or increased mutation rate) would be over-represented.

• I disagree with the conclusions drawn from Fig. 6A and 6B, specifically that “Pel production is activated by mutations leading to increased c-di-GMP” (lines 530-531). The data indicate that Pel contributes to the ability of the mutants to colonize the A-L interface but not that Pel production was impacted by the mutations. Direct evidence is needed to show that Pel production is increased in the mutants.

• Adjacent black bars makes it impossible to compare individual genes across species in Fig. 7.

• Has the whole genome sequence data been deposited into a public database?

**Have all data underlying the figures and results presented in the manuscript been provided?**

Reviewer #1: Yes

Reviewer #2: Yes

Reviewer #3: **No: **Fig 4B, Fig 6B missing underlying data

PLOS authors have the option to publish the peer review history of their article (what does this mean?). If published, this will include your full peer review and any attached files.

Reviewer #1: No

Reviewer #2: No

Reviewer #3: No

---

## [Decision Letter · Decision Letter 1]

16 Jul 2021

Dear Dr Lind,

We are pleased to inform you that your manuscript entitled "Forecasting of phenotypic and genetic outcomes of experimental evolution in Pseudomonas protegens" has been editorially accepted for publication in PLOS Genetics. Congratulations!

Yours sincerely,

Olivier Tenaillon

Guest Editor

PLOS Genetics

Kirsten Bomblies

Section Editor: Evolution

PLOS Genetics

Comments from the reviewers (if applicable):

We are happy to accept this revised version of the manusctript that has really improved over the last one and convinced the reviewers.

Reviewer's Responses to Questions

**Comments to the Authors:**

Reviewer #2: We thank the authors for their detailed and thoughtful response to our comments. Although we seem to differ with the authors with respect to our wish for a more principled, perhaps more course-grained predictive model, the response of the authors made us realize that this was perhaps too ambitious, and the more modest progress of their study is still informative and more than just a test of the reproducibility of previous findings. We particularly like the analyses and discussions of the possible reasons for variation in fitness effects and mutation rates of the adaptive mutations observed in the experiments with the new strain, which are potential leads to understanding their cause. We have no further comments or suggestions and look forward to seeing the future synthesis of all these details into a more generic predictive model.

**Have all data underlying the figures and results presented in the manuscript been provided?**

Reviewer #2: Yes

PLOS authors have the option to publish the peer review history of their article (what does this mean?). If published, this will include your full peer review and any attached files.

Reviewer #2: No

**Data Deposition**

http://datadryad.org/submit?journalID=pgenetics&manu=PGENETICS-D-21-00305R1

**Press Queries**

---

## [Editor Report · Acceptance letter]

30 Jul 2021

PGENETICS-D-21-00305R1 

Forecasting of phenotypic and genetic outcomes of experimental evolution in Pseudomonas protegens 

Dear Dr Lind, 

We are pleased to inform you that your manuscript entitled "Forecasting of phenotypic and genetic outcomes of experimental evolution in Pseudomonas protegens" has been formally accepted for publication in PLOS Genetics! Your manuscript is now with our production department and you will be notified of the publication date in due course.

With kind regards,

Andrea Szabo

PLOS Genetics

On behalf of:
